# The Role of Intestinal Cytochrome P450s in Vitamin D Metabolism

**DOI:** 10.3390/biom14060717

**Published:** 2024-06-17

**Authors:** Minori Uga, Ichiro Kaneko, Yuji Shiozaki, Megumi Koike, Naoko Tsugawa, Peter W. Jurutka, Ken-Ichi Miyamoto, Hiroko Segawa

**Affiliations:** 1Department of Applied Nutrition, Tokushima University Graduate School of Biomedical Sciences, Tokushima 770-8503, Japan; 2Research Institute for Food and Nutritional Sciences, School of Human Science and Environment, University of Hyogo, Hyogo 670-0092, Japan; 3Faculty of Nutrition, Kobe Gakuin University, Hyogo 651-2180, Japan; 4Mathematical and Natural Sciences, Arizona State University, Glendale, AZ 85306, USA; 5College of Medicine, The University of Arizona, Phoenix, AZ 85004, USA; 6Graduate School of Agriculture, Ryukoku University, Shiga 520-2194, Japan

**Keywords:** vitamin D, CYP3A, CYP24A1, 25OHD_3_, 1,25(OH)_2_D_3_

## Abstract

Vitamin D hydroxylation in the liver/kidney results in conversion to its physiologically active form of 1,25-dihydroxyvitamin D_3_ [1,25(OH)_2_D_3_]. 1,25(OH)_2_D_3_ controls gene expression through the nuclear vitamin D receptor (VDR) mainly expressed in intestinal epithelial cells. Cytochrome P450 (CYP) 24A1 is a catabolic enzyme expressed in the kidneys. Interestingly, a recently identified mutation in another CYP enzyme, CYP3A4 (gain-of-function), caused type III vitamin D-dependent rickets. CYP3A are also expressed in the intestine, but their hydroxylation activities towards vitamin D substrates are unknown. We evaluated CYP3A or CYP24A1 activities on vitamin D action in cultured cells. In addition, we examined the expression level and regulation of CYP enzymes in intestines from mice. The expression of CYP3A or CYP24A1 significantly reduced 1,25(OH)_2_D_3_-VDRE activity. Moreover, in mice, *Cyp24a1* mRNA was significantly induced by 1,25(OH)_2_D_3_ in the intestine, but a mature form (approximately 55 kDa protein) was also expressed in mitochondria and induced by 1,25(OH)_2_D_3_, and this mitochondrial enzyme appears to hydroxylate 25OHD_3_ to 24,25(OH)_2_D_3_. Thus, CYP3A or CYP24A1 could locally attenuate 25OHD_3_ or 1,25(OH)_2_D_3_ action, and we suggest the small intestine is both a vitamin D target tissue, as well as a newly recognized vitamin D-metabolizing tissue.

## 1. Introduction

Hypovitaminosis D is common in inflammatory bowel disease, especially in Crohn’s disease patients. These patients are at risk of developing secondary hyperparathyroidism and low bone mineral density [1]. The mechanism of intestinal vitamin D metabolism and its control of physiological vitamin D status and bone mineral density have been extensively studied [2,3].

Vitamin D is not only derived from food, but it is also synthesized from 7-dehydrocholesterol in the skin by ultraviolet rays. Vitamin D is converted to 25-hydroxyvitamin D_3_ [25OHD_3_] by liver 25-hydroxylase [Cytochrome P450 (CYP) 27A1, CYP2R1]. Blood 25OHD_3_ concentration is an index of vitamin D sufficiency, and correlations with bone mineral density and mortality risk have been reported [4,5,6]. 25OHD_3_ is converted to active vitamin D [1,25(OH)_2_D_3_] by a renal vitamin D-activating enzyme (CYP27B1). In the kidney, 1,25(OH)_2_D_3_ induces the expression of a vitamin D catabolic enzyme (CYP24A1), and this enzyme metabolizes 1,25(OH)_2_D_3_ to the inactive form by hydroxylation at the C24 position. The expression of these renal CYP enzymes is required to strictly maintain circulating 25OHD_3_ or 1,25(OH)_2_D_3_ levels [7,8]. 1,25(OH)_2_D_3_ binds to the vitamin D receptor (VDR) in target tissues and forms a heterodimer with the retinoid X receptor (RXR) to regulate the expression of vitamin D target genes such as *TRPV6*, *calbindin-9k*, etc., in the intestine through vitamin D response elements (VDREs) present in the genomic DNA [9,10,11].

All vitamin D-metabolizing enzymes are also classified as CYP enzymes [12]. All members of the CYP family are oxidoreductases with heme iron in the active center, and there are 57 molecular species in humans, 51 of which are localized in the endoplasmic reticulum and 6 of which are localized in the mitochondria [13,14]. CYP24A1 is thought to be localized in mitochondria and metabolizes 25OHD_3_ and 1,25(OH)_2_D_3_ to their inactive form by hydroxylating the 24 position in kidneys [15,16,17]. CYP24A1 has signal peptide at N-terminus (35 amino acids). Translated CYP24A1 (precursor, approximately 60 kDa) is transported to the mitochondria and becomes the mature form (55 kDa) with the signal peptide cleaved [15,18]. *CYP24A1* is also a vitamin D target gene and has several VDREs in the promoter and enhancer [8,9]. VDR expression is highest in the intestine, suggesting that the small intestine is a major vitamin D target organ [19,20].

Unlike CYP24A1, CYP3A4 is mainly expressed in liver and is a known xenobiotic enzyme [21]. Recently, it has been reported that CYP3A4 functions as a vitamin D catabolic enzyme that converts 25-hydroxyvitamin D_3_ to 4beta,25-dihydroxyvitamin D_3_ in vitro [22,23,24]. The human CYP3A family has four molecular species (*CYP3A4*, *CYP3A5*, *CYP3A7*, *CYP3A43*), and the amino acid sequence homology to CYP3A4 is as high as 84.3% (CYP3A5), 88.5% (CYP3A7), and 75.9% (CYP3A43). CYP3A4 is highly expressed in the liver and small intestine, CYP3A5 in the liver, and CYP3A7 in the fetal liver [25,26]. Furthermore, it was reported that *CYP3A4* gain-of-function mutations alter substrate affinity or substrate recognition, leading to type III vitamin D-dependent rickets (VDDR3) [27,28].

In the kidney, *CYP24A1* mRNA expression is mainly induced by fibroblast growth factor 23 or 1,25(OH)_2_D_3_, and its expression is suppressed by parathyroid hormone [29,30]. However, the mechanism of *CYP24A1* regulation in the small intestine is unclear.

Although the importance of the small intestine as a vitamin D target tissue has been recognized in numerous studies, the function and regulation of vitamin D metabolism in the small intestine is not well understood. In this study, we focused on the small intestine, which may be important as a vitamin D metabolic tissue, and we analyzed the expression and regulatory mechanisms of CYP3A and CYP24A1.

## 2. Materials and Methods

### 2.1. Cell Culture and Transfection

Human embryonic kidney (HEK) 293 cells were cultured in DMEM/F12 (D8062, Sigma, Burlington, MA, USA), and human colon cancer (HCT116) cells, and brush border-expressing subclone of Caco-2 (C2BBe1) cells were cultured in DMEM high glucose (D6429, Sigma, Burlington, MA, USA). Cell cultures were maintained in media supplemented with 10% fetal bovine serum (FBS) and penicillin/streptomycin at 37 °C with 5% CO_2_ and were passaged with 0.05% trypsin/EDTA every 3 days. Transfections for protein expression were performed using polyethylenimine Max (Polysciences, Warrington, PA, USA) in HEK293 cells and with Lipofectamine2000 (Thermo Fisher Scientific, Waltham, MA, USA) in C2BBe1 cells. Forty-eight hours after transfection, HEK293 cells were lysed with 20 mM HEPES-KOH (pH7.4)/150 mM NaCl/1% Triton X-100/1% sodium dodecyl sulfate (SDS).

### 2.2. Plasmid Constructs

Full-length human CYP3A4, 5, 7, CYP24A1, and mouse Cyp24a1 were fused to the C-terminus of the FLAG peptide by inserting them into the pCMV-Tag2B vector (Stratagene, San Diego, CA, USA). Full-length human CYP24A1 and mouse Cyp24a1 were fused to the N-terminus of the FLAG peptide by inserting them into a pCMV-Tag4A vector (Stratagene, San Diego, CA, USA). N-terminal deleted (N-del) human CYP24A and N-del mouse Cyp24a1, which were deleted at the N-terminus (35 amino acids), were fused to the C-terminal FLAG tag peptide by inserting them into a pCMV-Tag2A vector (Stratagene, San Diego, CA, USA). The primers for cloning are shown in Table 1.

### 2.3. Luciferin-IPA Assay

Luciferin-IPA assays (Promega, Madison, WI, USA) were performed following the manufacturer’s protocol. Luciferin-IPA is converted to Luciferin by hydroxylation via CYP3A4 and becomes a luminescent substrate for Luciferase [31]. Transfection efficiency was corrected for renilla luciferase activity.

### 2.4. Dual Luciferase Reporter Assay

HCT116 or HEK293 cells in 24-well plates were transfected with a vitamin D response element luciferase reporter vector (VDRE-Luc, 150 ng); pSG5-VDR (40 ng); pRL-Null (Renilla luciferase, 10 ng); pCMV-Tag2B-human CYP3A4 or CYP3A5, CYP3A7, and CYP24A1 (50 ng) using ViaFect Transfection Reagent (Promega, Madison, WI, USA). 1,25(OH)_2_D_3_-dependent VDR-RXR heterodimerization was analyzed by using the mammalian two-hybrid assay. In this assay, HCT116 cells were transfected with pFR-Luc (150 ng); GAL4 DNA-binding domain-human VDR (25 ng); activation domain-human retinoid X receptor (25 ng); pRL-Null (10 ng); pCMV-Tag2B-human CYP3A4 or 5, 7 CYP24A1 (50 ng). Twenty-four hours after transfection, cells were treated with 1,25(OH)_2_D_3_, 25OHD_3_, or ethanol vehicle for twenty-four hours. The cells were lysed in 1×passive lysis buffer (Promega, Madison, WI, USA), and firefly luciferase and renilla luciferase activity was measured using a Dual-Luciferase Reporter Assay Kit (Promega, Madison, WI, USA).

### 2.5. Animal Studies

All mice were generated with a C57BL/6J background, fed a standard diet (Oriental Yeast Co., Ltd., Tokyo, Japan) and maintained on a 12-h light/dark cycle controlled under specific-pathogen-free conditions. Intestine-specific *VDR* knockout mice (VDR-vKO) were generated by crossing *VDR*^lox/lox^ mice (kindly provided by Dr. Kato, Iryo Sosei University) with transgenic mice expressing Cre recombinase under control of the villin promoter (Jackson Lab, Bar Harbor, ME, USA). VDR^lox/lox^ mice were used as control mice [32,33].

The male mice (10-week of age) were orally administrated 1,25(OH)_2_D_3_ (Sigma, Burlington, MA, USA) solved in MCT oil (10 ng/g body weight). All samples were harvested after 6 h [34]. The upper half of the small intestine is designated as the proximal intestine, and the lower half of the small intestine is designated as the distal intestine.

### 2.6. RNA Isolation and Quantitative Reverse Transcription—PCR Analysis

Total RNA was isolated from tissues using ISOGEN (NIPPON GENE Co., Ltd., Tokyo, Japan) according to the manufacturer’s instructions. Next, cDNA was synthesized from 1 µg of DNase I-treated total RNA using Moloney Murine Leukemia virus reverse transcriptase (Thermo Fisher Scientific, Waltham, MA, USA) and oligo(dT) primers. Quantitative PCR was performed using StepOnePlus Real-Time PCR Systems (Applied Biosystems, Waltham, MA, USA). The reaction mixture contained 10 µL of TB Green Premix ExTaq II (TaKaRa, Shiga, Japan), ROX Reference Dye II, and specific primers shown in Table 2.

### 2.7. Protein Preparation

Intestine and kidney were homogenized in a 20 mM HEPES-KOH (pH7.4)/1 mM EDTA/1% SDS-containing protease inhibitor cocktail (Roche, Basel, Switzerland). After centrifugation (12,000 rpm), the supernatants were collected as whole homogenate samples.

Intestine and kidney were homogenized in 100 mM KCl/50 mM Tris-HCl/2 mM EDTA and centrifuged at 500× *g*. Next, the supernatants were collected and centrifuged (10,500× *g*). The mitochondrial pellets were lysed in 20 mM HEPES/150 mM NaCl/1% Triton X-100/1% SDS. The supernatants were collected as cytoplasmic lysates.

### 2.8. Western Blotting

Proteins were prepared in SDS-sample buffer in the presence of 2-mercaptoethanol and separated by sodium dodecyl sulfate–polyacrylamide gel electrophoresis. The separated proteins were transferred by electrophoresis to Immobilon-P PVDF (Millipore, Billerica, MA, USA) and incubated with one of the following primary antibodies overnight: anti-CYP3A (B-3 for CYP3A4 or 5, 7: P-6 for Cyp3a11 or 13: SantaCruz, Dallas, TX, USA), anti-Cyp24a1 (GTX105884: GeneTex, Irvine, CA, USA), anti-VDR (D6: SantaCruz, Dallas, TX, USA), anti-Tom20 (Proteintech, Rosemont, IL, USA), anti-FLAG-M2 (Sigma, Burlington, MA, USA), anti-tubulin (A6: SantaCruz, Dallas, TX, USA) or anti-actin (Millipore, Billerica, MA, USA). Horseradish peroxidase-conjugated anti-rabbit or anti-mouse IgG was used as the secondary antibody (Jackson Immuno Research Laboratories, Inc., West Grove, PA, USA), and signals were detected using Immobilon Western (Millipore, Billerica, MA, USA) on an Amersham Imager 600 (Cytiva, Marlborough, MA, USA).

### 2.9. Measurement of Plasma 25OHD_3_ and 24,25(OH)_2_D_3_ Concentrations by LC/MS/MS Analysis

Plasma concentrations of 25OHD_3_ and 24,25(OH)_2_D_3_ were measured by using a modified method of LC-APCI-MS/MS [35]. Internal standard *d*_6_-25OHD_3_ (10 ng/50 μL) and 0.18 mL of water were added to plasma (0.02 mL), followed by acetonitrile (1.0 mL) for protein removal. The mixture was shaken and centrifuged at 3000 rpm for 10 min. Supernatant was taken and evaporated. The resultant residue was dissolved with 1.2 mL of ethyl acetate, and 0.6 mL of distilled water was added. After centrifugation at 3000 rpm for 10 min, the ethyl acetate phase was taken and evaporated. Extracted vitamin D metabolites were derivatized by 4-[2-(6,7-dimethoxy-4-methyl-3-oxo-3,4-dihydroquinoxalyl)ethyl]-1,2,4-triazoline-3,5-dione (DMEQ-TAD). Apparatus of LC-APCI-MS/MS (QTRAP^®^ 4500 LC-MS/MS System, AB Sciex Pte. Ltd. Framingham, MA, USA) was used. Analytical column: CAPCELL PAK C18 UG120, 5 μm; (4.6 I.D. × 250 mm) (SHISEIDO, Tokyo, Japan). Mobile phase: acetonitrile. Flow rate: 1.0 mL/min. All MS data were collected in the positive ion mode and quantitative analysis was carried out using MS/MS-MRM of the precursor/product ion for DMEQ-TAD-25OHD3 (*m*/*z*: 746.5/468.1), DMEQ-TAD-24,25(OH)_2_D_3_ (*m*/*z*: 762.5/468.0), and DMEQ-TAD-*d*_6_-25OHD_3_ (*m*/*z*: 752.5/468.1) with a dwell time of 300 ms.

### 2.10. Statistics

Data are expressed as means ± SE. Statistical differences between two groups were determined using a two-tailed Student’s *t*-test. Differences among multiple groups were analyzed by ANOVA, and then differences among means were analyzed using Tukey–Kramer comparison tests. A *p*-value of less than 0.05 was considered significant. Graphs were drawn using Prism9.

## 3. Results

### 3.1. Effects of CYP3A or CYP24A1 on 1,25(OH)_2_D_3_ Activity in Human Colon Cells

Plasmids expressing the indicated proteins with a FLAG-tag fused to the N-terminal side of human CYP3A4, 3A5, 3A7, and CYP24A1 were prepared and expressed in HEK293 cells. Anti-FLAG detected each of the CYP enzymes at comparable levels. Anti-CYP3A (B-3) cross-reacted with the isoforms CYP3A4, CYP3A5, and CYP3A7. Anti-CYP24A1 (which recognizes an epitope of the amino acids 241–488 in the human protein, 514aa) detected two bands (see arrows) of approximately 60 kDa (precursor) and a less abundant band at 55 kDa (mature form) (Figure 1a).

We next performed a Luciferin-IPA assay to assess whether our original CYP3A4 plasmid is functional. The transfection of HEK293 cells with the CYP3A4 expression plasmid strongly induced hydroxylation activity in a Luciferin-IPA assay, and this activity was potently inhibited by ketoconazole, a selective inhibitor of CYP3A4 (Figure 1b).

VDRE-mediated transcriptional activity was measured using a VDRE-linked reporter assay in the presence of vitamin D. In HCT116 cells, addition of 100 nM 1,25(OH)_2_D_3_ increased VDRE-mediated transcriptional activity (set to 100%), but this activity decreased by the co-transfection of plasmids encoding CYP3A4, CYP3A5, CYP3A7, or CYP24A1 (Figure 1c). The formation of the VDR-RXR hetero-complex in a mammalian two-hybrid assay was also increased by 10 nM 1,25(OH)_2_D_3_ and was statistically significantly suppressed by the expression of CYP3A4, CYP3A5, and CYP3A7 (Figure 1d). Utilizing qPCR, we found that endogenous *TRPV6* mRNA expression in C2BBe1 cells was increased by 10 nM 1,25(OH)_2_D_3_ administration, followed by the significant suppression of *TRPV6* in the presence of CYP3A4 and CYP24A1 expression plasmids (Figure 1e).

### 3.2. Functional Effects of CYP24A1 on 1,25(OH)_2_D_3_ Activity

Plasmids coding for proteins in which the FLAG-tag was fused to the N-terminal or C-terminal end of human full-length CYP24A1 or mouse full-length Cyp24a1 were overexpressed in HEK293 cells. Both human CYP24A1 and mouse Cyp24a1 fused with a FLAG-tag at the N-terminal were detected as one band with an anti-FLAG antibody and detected as two bands with an anti-CYP24A1 antibody. When the FLAG-tag was fused to the C-terminus, the CYP24A1 was detected as two bands by both the anti-FLAG antibody and anti-CYP24A1 antibody (Figure 2a). These data show the CYP24A1 antibody can recognize precursor or mature forms in both humans and mice.

As a model for loss of the CYP24A1 signal peptide (loss of transport to mitochondria), an N-del CYP24A1 expression plasmid was overexpressed in HEK293 cells. N-del CYP24A1 was detected as a smaller-molecular-weight band compared to full-length CYP24A1. In addition, N-del CYP24A1 was detected as a single band with both an anti-FLAG antibody and anti-CYP24A1 antibody, suggesting that no further truncation occurred in the N-del protein (Figure 2b).

1,25(OH)_2_D_3_ activity was evaluated using a VDRE-mediated transcriptional reporter assay. VDRE-mediated transcriptional activity was increased by the administration of 1,25(OH)_2_D_3_ and was suppressed by the overexpression of full-length CYP24A1. Parts of human CYP24A1 or mouse Cyp24a1 expressed mature forms. The mature CYP24A1 suppressed the 1,25(OH)_2_D_3_-mediated activity (Figure 2c). In contrast, the expression of either human or mouse N-del CYP24A1 did not suppress 1,25(OH)_2_D_3_-mediated activity because the N-del CYP24A1 lost mitochondrial transport (Figure 2c).

### 3.3. Tissue Expression of Cyp3a and Cyp24a1 in Mice

Mouse *Cyp3a11* and *Cyp3a13* genes are orthologs of human *CYP3A4*. *Cyp3a11* mRNA is mainly expressed in the liver and intestine, while *Cyp3a13* mRNA is predominantly expressed in the intestine (Figure 3a). Intestinal *Cyp24a1* mRNA was minimal and only significantly expressed in kidneys (Figure 3b). We also tested and confirmed that anti-CYP3A antibody (P-6) cross-reacts with mouse Cyp3a11 and Cyp3a13 (Appendix A). Western blots confirmed that total Cyp3a protein is detected in the proximal intestine but is highest in the liver (Figure 3c).

### 3.4. 1,25(OH)_2_D_3_-Dependent Regulation of Cyp3a or Cyp24a1 in the Mouse Intestine

*Cyp3a11* and *Cyp3a13* mRNA expression was not induced in the small intestine, kidney, and liver of wild-type mice treated with 1,25(OH)_2_D_3_ (Figure 4a). However, *Cyp24a1* mRNA expression was significantly increased in the small intestine and kidney (Figure 4b). Importantly, the induction of *Cyp24a1* mRNA expression in the small intestine was remarkable (374,156-fold in proximal and 335,318-fold in distal), along with a 4.7-fold induction in kidney, all compared to the vehicle. Moreover, when Cyp24a1 protein expression was assessed by Western blotting analysis, a band corresponding to mature Cyp24a1 protein (55 kDa) was detected in mice treated with 1,25(OH)_2_D_3_ (Figure 4c). We subsequently confirmed that this band was not derived from a splicing variant but was due to post-translational processing (Appendix A).

We investigated the intracellular localization of mature Cyp24a1 proteins in the small intestine. Mature Cyp24a1 was only exclusively localized in small intestine mitochondria in 1,25(OH)_2_D_3_-treated mice (Figure 4d).

### 3.5. 1,25(OH)_2_D_3_-Dependent Regulation of Cyp24a1 in Intestine-Specific VDR-KO Mice

We next measured the expression of *Cyp24a1* mRNA in the intestine and kidney of both control and VDR-vKO (intestine-specific KO) mice (Figure 5a). As expected, 1,25(OH)_2_D_3_ significantly induced *Cyp24a1* mRNA in the intestine of control mice, but there was no change in *Cyp24a1* expression in VDR-vKO mice (Figure 5a). In contrast, 1,25(OH)_2_D_3_ induced *Cyp24a1* mRNA in the kidneys of control mice and did so even more effectively in VDR-vKO mice. Mature intestinal Cyp24a1 protein was observed in control mice treated with 1,25(OH)_2_D_3_ but was undetected in 1,25(OH)_2_D_3_-treated VDR-vKO mice (Figure 5b). These novel results imply that intestinal mature Cyp24a1 is a VDR-dependent protein that is dramatically induced by 1,25(OH)_2_D_3_.

In order to determine if there is a potential functional effect of the intestinal mature Cyp24a1 protein on vitamin D metabolism, plasma 25OHD_3_ and 24,25(OH)_2_D_3_ levels were measured. There was no change in plasma 25OHD_3_ concentrations with 1,25(OH)_2_D_3_ administration in control or VDR-vKO mice. However, the plasma 24,25(OH)_2_D_3_ levels in 1,25(OH)_2_D_3_-treated control mice significantly increased compared to vehicle-treated control mice. In VDR-vKO, the increase in plasma 24,25(OH)_2_D_3_ with 1,25(OH)_2_D_3_ administration was not observed (Figure 5c). These results suggest the intriguing possibility that intestinal mature Cyp24a1 might function to hydroxylate 25OHD_3_ to 24,25(OH)_2_D_3_.

## 4. Discussion

Vitamin D is hydroxylated in the liver and kidney and converted to 1,25(OH)_2_D_3_ as the final active hormonal metabolite. 1,25(OH)_2_D_3_ induces renal CYP24A1 expression and is thus metabolized to its inactive form to maintain circulating 1,25(OH)_2_D_3_ levels (negative feedback). Using in vitro cell culture models, it has been demonstrated that CYP3A4 can hydroxylate 25OHD_3_ or 1,25(OH)_2_D_3_ and is a candidate for vitamin D catabolic enzymes [23,36,37]. Genetic mutations in *CYP3A4*, as well as the CYP3A4 drug-induced excessive catabolism of vitamin D, lead to metabolic bone disorders [22,27,38]. However, the regulatory mechanism of vitamin D metabolic enzymes, especially in the small intestine (which is important as a vitamin D target organ), remains unclear. Therefore, the purpose of this study was to elucidate the mechanism of vitamin D metabolism and regulation by CYP3A and CYP24A1 in the small intestine.

First, we investigated the effects of CYP3A on 1,25(OH)_2_D_3_-mediated transcriptional activity and endogenous target gene (*TRPV6*) expression using cultured cells derived from human intestinal epithelium and found that CYP3A acts to metabolize and inactivate 1,25(OH)_2_D_3_ as well as CYP24A1 in the small intestine in vitro. Therefore, we examined *Cyp3a* and *Cyp24a1* expression levels in mouse small intestine and found that Cyp3a was highly expressed in the upper small intestine in vivo. Next, we examined the regulation of Cyp expression and found that 1,25(OH)_2_D_3_ did not induce *Cyp3a* expression in the small intestines of mice. While human *CYP3A* are induced by 1,25(OH)_2_D_3_ through a classical VDRE in the human promoter region [39], mouse *Cyp3a11/13* expression was not induced by 1,25D in this study. Several mouse Cyp3a isoenzymes could be target genes for vitamin D in the intestine under certain conditions [40]. Moreover, a VDRE has been identified in the mouse *Cyp3a11* promoter, and *Cyp3a11* was induced by 1,25D after 3 days of hormone treatment in the intestine or liver of WT mice [41]. Thus, it is possible that the timeframe (or other conditions) may be different for *Cyp3a* induction in mice compared to the rapid induction of *Cyp24a1*. In fact, we found that *Cyp24a1* mRNA expression in mouse small intestines was strongly increased by 1,25(OH)_2_D_3_, even more so than that observed in mouse kidneys. It was characterized by the poor expression of *Cyp24a1* in the intestine but displayed a quick response compared to *Cyp3a*.

A previous report showed that CYP24A1 requires adrenodoxin and NADPH-adrenodoxin reductase as a mitochondrial electron transport system to hydroxylate 1,25(OH)_2_D_3_ or 25OHD_3_ [42]. In the intestines of mice, precursor Cyp24a1 protein with an estimated molecular weight of about 60 kDa was not detected, but a 1,25(OH)_2_D_3_-dependent mature Cyp24a1 protein of about 55 kDa was detected in the mitochondria. Normally, mitochondrial proteins are synthesized on the ribosome, then transported into the mitochondria by recognition of the mitochondrial signal peptide via the translocase of outer membrane 20 (Tom20) present in the mitochondrial outer membrane. The signal peptide is cleaved and removed after crossing the mitochondrial membrane [43]. Therefore, Cyp24a1 55kDa protein detected in mouse small intestines is presumed to be a mitochondrial signal peptide cleavage fragment (consisting of 35 amino acids at the N-terminus of full-length Cyp24a1) with hydroxylation activity. Next, we analyzed the function of mature Cyp24a1 in vivo. The results using VDR-vKO mice showed that plasma 24,25(OH)_2_D_3_ levels were significantly increased in the presence of mature Cyp24a1 in control mice, but there was no change in VDR-vKO mice.

It has been reported that human and rat *CYP24A1* have a splicing variant lacking exon 1 and exon 2 (CYP24A1-SV, estimated molecular weight of about 40 kDa) in the kidney cytoplasm, which is increased in a diabetic rat model [44,45]. In this study, we investigated the induction of mouse *Cyp24a1-SV* mRNA expression by 1,25(OH)_2_D_3_, but no response was observed (Appendix A). We also detected a GFP-tagged N-fragment of CYP24A1 (Appendix A). These results indicate that mature Cyp24a1 protein is due to protein processing (signal peptide removal) and not derived from a *Cyp24a1-SV*gene.

CYP3A has been suggested to locally regulate vitamin D action in the small intestine. The *CYP3A4* gene is induced by VDR (in humans); pregnane X receptor (PXR); and constitutive androstane receptor (CAR) ligands such as rifampicin (antituberculous drug), carbamazepine, phenobarbital (antiepileptic drug), pioglitazone (insulin sensitizer), and St. John’s wort (Hypericum perforatum, antidepressant) [46,47,48,49]. Rifampicin administration to patients with idiopathic infantile hypercalcemia (IIH) due to loss-of-function mutations in *CYP24A1* induces *CYP3A4*, suggesting an alternative pathway for vitamin D inactivation [50]. In addition, methotrexate chemotherapy (which is commonly used in childhood oncology) leads to bone loss and decreasing serum 25OHD_3_ along with altered intestinal *Cyp24a1* [51]. During drug therapy in clinical practice, multiple drugs are frequently used, and adverse drug interactions can sometimes occur. For osteoporosis or chronic kidney disease patients, an active vitamin D derivative could induce *CYP3A4* and *CYP24A1* in the intestines. This may be attenuated by concomitant use with CYP inducers.

This study suggests the importance of CYP3A action and CYP24A1 regulation in vitamin D inactivation in the small intestine. Elucidation of the effects of *CYP3A4* or *CYP24A1* inducers on vitamin D action, and the mechanism of CYP24A1 processing in the intestine, will lead to improvements in vitamin D repletion. Until now, the liver and kidney were thought to be the major vitamin D-metabolizing tissues. Our current results suggest that the small intestine is both a vitamin D target tissue, as well as a newly recognized vitamin D-metabolizing tissue, and this is the first report of the mature CYP24A1 protein in intestine. Our next challenge is to clarify chronic effects and physiological roles in this pathway.

## Figures and Tables

**Figure 1 biomolecules-14-00717-f001:**
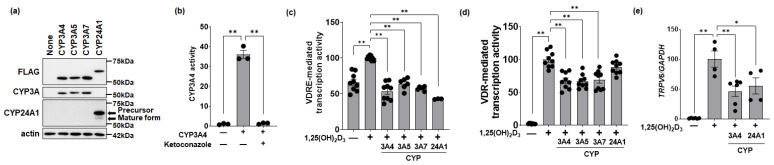
Human CYP3A or CYP24A1 activity. (**a**) Total cell lysate (20 µg/lane) from HEK293 transfected with FLAG-tagged CYP3A or CYP24A1 at the N-terminus was separated on an 8% SDS-PAGE gel and probed in a Western blot with the indicated antibodies. (**b**) CYP3A4 activity was confirmed with the Luciferin-IPA assay in HEK293. Ketoconazole (10 µM) was added as a selective inhibitor for CYP3A4 (*n* = 3). (**c**) A luciferase plasmid containing the VDRE from rat osteocalcin was used to measure transcriptional activity by 1,25(OH)_2_D_3_ (100 nM) in HCT116 (*n* = 3–9). (**d**) A mammalian two-hybrid assay (with VDR and RXR bait/prey) was performed with 1,25(OH)_2_D_3_ (10 nM) in HCT116 (*n* = 9). (**e**) The expression of a classical vitamin D target gene (*TRPV6*) was assessed via qPCR in C2BBe1 gut cells administrated 1,25(OH)_2_D_3_ (10 nM). *GAPDH* was used as an internal control. Vehicle-treated cells were set to 1.0 (mean ± SE; *n* = 4–6; * *p* < 0.05, ** *p* < 0.01). Original images can be found in Appendix A.

**Figure 2 biomolecules-14-00717-f002:**
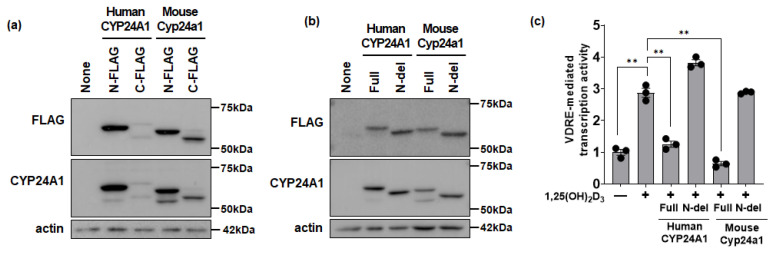
N-del CYP24A1 does not inactivate transcriptional activity by 1,25(OH)_2_D_3_. (**a**) Analysis of total cellular lysates (20 µg/lane) from HEK293 cells transfected with N- and C-FLAG-tagged human CYP24A1 or mouse Cyp24a1 via Western blotting on 8% SDS-PAGE gel. (**b**) Western blot of total cellular lysate (20 µg/lane) from HEK293 cells transfected with FLAG-tagged human CYP24A1 (full-length or N-del) or mouse Cyp24a1 (full-length or N-del) and separated by 8% SDS-PAGE gel. (**c**) Luciferase plasmid containing the VDRE from the rat osteocalcin gene was used to measure transcriptional activation by 1,25(OH)_2_D_3_ (10 nM) in HEK293 cells in the presence of expressed full-length or N-del CYP24A1 expression vectors. (mean ± SE; *n* = 3; ** *p* < 0.01). Original images can be found in Appendix A.

**Figure 3 biomolecules-14-00717-f003:**
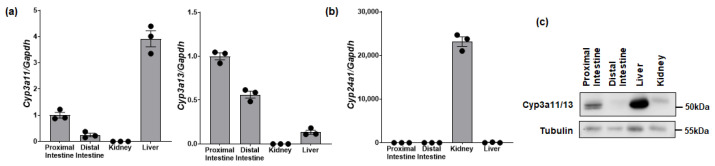
The expression of mouse *Cyp3a* or *Cyp24a1* in intestine. (**a**) *Cyp3a11* or *Cyp3a13* mRNA expression and (**b**) *Cyp24a1* mRNA expression in various tissues extracted from wild-type mice (male, 10 weeks of age). The primers for real-time PCR used the *Cyp24a1* exon 6–7 (Table 2, Appendix A). *Gapdh* was used as the internal control. The expression in the proximal intestine was set to 1.0. (mean ± SE; *n* = 3). (**c**) Western analysis of Cyp3a11/13 protein expression. The whole homogenate (20 µg/lane) was separated by 8% SDS-PAGE gel (*n* = 3). Original images can be found in Appendix A.

**Figure 4 biomolecules-14-00717-f004:**
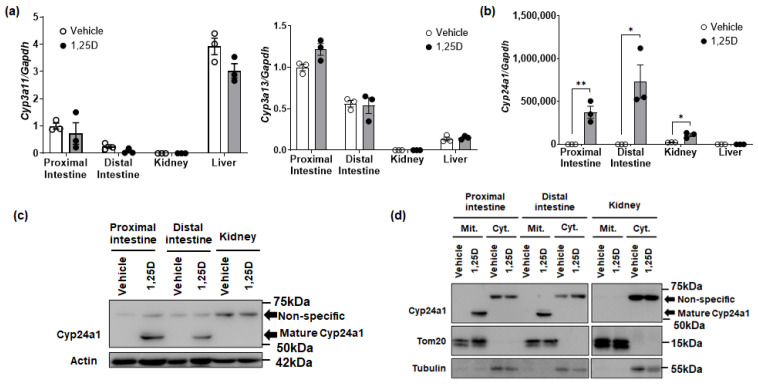
*Cyp3a* or *Cyp24a1* induction by 1,25(OH)_2_D_3_ in mice. Wild-type mice (male, 10 weeks of age) were treated with 1,25(OH)_2_D_3_ (10 ng/g BW) and assessed for (**a**) *Cyp3a11* or *Cyp3a13* mRNA expression or (**b**) *Cyp24a1* mRNA expression in various tissues. The primers for real-time PCR used the *Cyp24a1* exon 6–7 (Table 2, Appendix A). *Gapdh* was used as an internal control. The expression in the proximal intestine administrated by the vehicle was set to 1.0 (mean ± SE; *n* = 3; * *p* < 0.05, ** *p* < 0.01). (**c**) Mature Cyp24a1 protein induction by 1,25(OH)_2_D_3_. The whole-cell homogenate (20 µg/lane) was separated by 8% SDS-PAGE gel (*n* = 3). (**d**) Mature Cyp24a1 protein expression in mitochondria or cytoplasm. The mitochondria (Mit) or cytoplasmic (Cyt) homogenates (20 µg/lane) were separated by 8% SDS-PAGE gel. Tom20 is a marker of mitochondria, and tubulin is a marker of cytoplasm (*n* = 3). Original images can be found in Appendix A.

**Figure 5 biomolecules-14-00717-f005:**
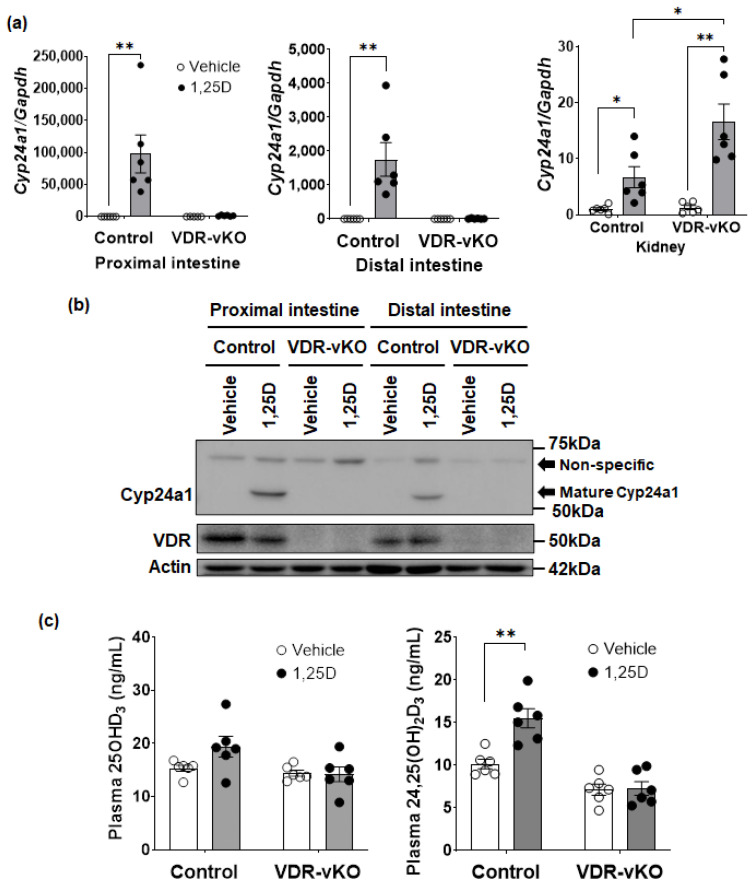
Mature Cyp24a1 in intestine affects circulating 24,25(OH)_2_D_3_ levels. VDR^flox^ mice (control) or VDR-vKO mice (male, 10 week of age) were treated with 1,25(OH)_2_D_3_ (10 ng/g BW). (**a**) The assessment of *Cyp24a1* mRNA induction. The primers for real-time PCR used *Cyp24a1* exon 6–7 (Table 2, Appendix A). *Gapdh* was used as an internal control. The expression in the proximal intestine administrated by the vehicle in control mice was set to 1.0 (mean ± SE; *n* = 6; * *p* < 0.05, ** *p* < 0.01). (**b**) Cyp24a1 protein induction by 1,25(OH)_2_D_3_ (10 ng/g BW) in VDR-vKO mice using Western blotting. The whole-cell homogenate (20 µg/lane) was separated by 8% SDS-PAGE gel (*n* = 3). (**c**) Circulating vitamin D metabolites 25OHD_3_ or 24,25(OH)_2_D_3_ were measured by LC-MS/MS (Mean ± SE; *n* = 6; ** *p* < 0.01). Original images can be found in Appendix A.

**Table 1 biomolecules-14-00717-t001:** Primers for cloning.

Gene	Forward (5′→3′)	Reverse (5′→3′)
Human *CYP3A4*	GCGCAGATCTATGGCTCTCATCCCAGACTTG	GCGCCTCGAGTCAGGCTCCACTTACGGTGCC
Human *CYP3A5*	GCGCAGATCTATGGACCTCATCCCAAATTTG	GCGCCTCGAGTCATTCTCCACTTAGGGTTCC
Human *CYP3A7*	GCGCAGATCTATGGATCTCATCCCAAACTTG	GCGCCTCGAGTCAGGCTCCACTTACGGTCTC
Human full-length *CYP24A1* N-FLAG	GCGCGAATTCATGAGCTCCCCCATCAGCAAGAG	GCGCGTCGACTTATCGCTGGCAAAACGCGATGG
Mouse full-length *Cyp24a1* N-FLAG	GCGCGAATTCATGAGCTGCCCCATTGACAAAAG	GCGCGTCGACCTACCGTGGACAGAACGCAATGG
Human full-length *CYP24A1* C-FLAG	GCGCGAATTCATGAGCTCCCCCATCAGCAAGAG	GCGCGTCGACTCGCTGGCAAAACGCGATGGGG
Mouse full-length *Cyp24a1* C-FLAG	GCGCGAATTCATGAGCTGCCCCATTGACAAAAG	GCGCGTCGACCCGTGGACAGAACGCAATGGGC
Human N-del *CYP24A1*	GCGCGAATTCCCTCAGCCGCGAGAGGTGCC	GCGCGTCGACTTATCGCTGGCAAAACGCGATGG
Mouse N-del *Cyp24a1*	GCGCGAATTCCGTGCGCCAAAAGAGGTGCC	GCGCGTCGACCTACCGTGGACAGAACGCAATGG

**Table 2 biomolecules-14-00717-t002:** Primers for real-time PCR.

Gene	Forward (5′→3′)	Reverse (5′→3′)
*GAPDH*	CTGCACCACCAACTGCTTAGC	CATCCACAGTCTTCTGGGTG
Human *TRPV6*	AAGCCTACATGACCCCTAAG	CCCATTCTGAAGATGTCTGG
Mouse *Cyp3a11*	TGGACAGAATGAAGGAAAGCC	GGCTTTATGAGAGACTTTGTC
Mouse *Cyp3a13*	GATGAAATTGATGCGGCTCTG	TCTCAAGTCTTCCAGCGATTG
Mouse *Cyp24a1* exon 6–7	TGGGAAGATGATGGTGACCC	TCGATGCAGGGCTTGACTG
Mouse *Cyp24a1* exon 1	TACTGCTCCTCGAGTGTCAC	CTTGGATGTCACGGACCTTG
Mouse *Cyp24a1* intron 2—exon 3	TAAGCATACCCCTTCTCTGC	CAGCTTCATGATTTCCACGG

## Data Availability

The datasets generated or analyzed during the current study are available from the corresponding author (I.K.) upon reasonable request.

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
