# Peer review of "The Role of Intestinal Cytochrome P450s in Vitamin D Metabolism"

_biomolecules, 2024, doi:10.3390/biom14060717_

Round 1

Reviewer 1 Report (New Reviewer)

Comments and Suggestions for Authors

The authors look at the role of two main P450 family, CYP24 and CYP3As in the metabolism of vitamin D in the intestine.

Background:

CYP24 and CYP3As are known to metabolize vitamin D (25-hydroxy and 1,25-dihydroxyvitamin D) to corresponding inactive compounds. CYP24 is mainly express in the kidney and CYP3A are mainly express in the liver and small intestine. To date not much is known about vitamin D metabolism in the intestine.

Results:

3.1: The author showed in section 3.1 that using transformed cells with CYP24 or CYP3As can block transcription of vitamin D regulated genes by metabolizing 1,25(OH)2D3.

A-Comments: It would have been very informative to characterize the structure of generated 1,25(OH)2D3 metabolites and compare their rate of formation in these systems to compare the activity of CYP3A4, 3A5, 3A7 and CYP24.

3.2:  In section 3.2 authors looked at the effect of CYP24A1 truncation on vitamin D metabolism:

B- CommentsMore details on the assay using a VDRE-mediated transcriptional reporter assay should be given to help the reader as it looks like activity was specifically measure in the mitochondria.

3.3: This section describes Cyp3a or Cyp24a1 expression in mouse intestine. CYP24 is poorly expressed in the intestine. CYP3A11 and 13 are both expressed in the intestine.

3.4: Here the authors showed that CYP3A11 and 13 are not induced by 1,25(OH)2D3 in all tested organs including intestine. In the discussion, “Next, we examined the regulation of Cyp expression, and found that 1,25(OH)2D3 did not induce Cyp3a expression in the small intestine of mice. … and Cyp3a11 was induced by 1,25D after 3 days of hormone treatment in intestine or liver of WT mice [40]” what is this suggesting? The authors should clarify the intent of this analysis and make appropriate suggestions as to its relevance.

However, CYP24, the spliced form (mature) was greatly induced based on the fact that untreated animals showed no or very low levels when untreated. 

C-CommentsIf CYP3As are involved in vitD3 regulation, to what extent do they play a role compared to CYP24.  Please comment on relevance.

3.5 And finally in this section author the authors used intestine specific VDR-KO mice to provide data on the mechanism of CYP24 regulation. Intestinal CYP24 (mature form) is regulated by 1,25(OH)2D3 through VDR as no upregulation is observed in the intestine specific VDR-KO animals. The authors also looked at the effect on 25(OH)D3 and 24,25(OH)2D3 plasma levels in normal or KO mice after 1,25(OH)2D3 treatment. No change in 25(OH)D3 was observed in KO animals. However, 24,25(OH)2D3 level was increased in normal mice but not in KO.

Comment: How do you explain that CYP24 in upregulated in mouse kidney (4.7-fold) after 1,25(OH)2D3 treatment and no increase of plasma 24,25(OH)2D3 in intestinal specific VDR KO animals.  The authors may wish to consider the role of FGF23 in this instance.

General comments:

1.     CYP3As are able to metabolize 25(OH)D3 to 4,25(OH)2D3 and 1,25(OH)2D3 to mainly 1,23,25(OH)3D3 but the authors did not look at these metabolites (or did not present data). These data would help to understand the difference between these isoforms for vitamin D metabolism.

2.     The fact that CYP3As are not regulated in the mouse by 1,25(OH)2D3 seems to suggest that mice might not be a good model to study CYP3A vitamin D metabolism in the intestine.

3.     It would also be interesting to correlate/look at VDR and CYP regulation in the intestine following treatment with 1,25(OH)2D3.

4.     To better understand the protein size on figures 3, 4 and 5, more molecular weight markers should be shown.

5.     In the discussion, “A previous report showed that CYP24A1 is necessary for adrenodoxin and NADPH adrenodoxin reductase as mitochondrial electron transport system to hydroxylate 1,25(OH)2D3 or 25OHD3 [41]” should read “A previous report showed that CYP24A1 requires adrenodoxin….

6.     In the introduction, a short review on CYP24 maturation (splicing) would be helpful for further reading.

7.     Line 186 in Materials and Methods, is pure acetonitrile used as mobile phase (no water, acid…)?

8.     Line 178 in Materials and Methods, volume of acetonitrile added for protein removal is missing.

9.     The primers for real time PCR should be stated as Table 2 (line 141)

Comments on the Quality of English Language

Quality of English is adequate for this publication.

Author Response

Reviewer 2 Report (New Reviewer)

Comments and Suggestions for Authors

Dear Authors,

Major concerns,

1.       Please note that this intriguing finding concerning the role of CYP3As in vitamin D metabolism is not supported by detection of metabolites, thus is just a concept, based on previously published data. Please, mention this.

2.       I understand that use of CYP24A1 expression for VDR activation assay might be problematic in a case of cells transformed with CYP24A1 construct, however in case of other CYPs it could be very sensitive test.

3.       Figure 2 Please provide evidence that CYP24A1 with N-Del is really not colocalized with mitochondria. Confocal microscopy or western blot (mitochondrial vs cytoplasmic fraction with appropriate controls). Similarly as on Figure 4.

4.       Please note that your interpretation of the data makes sense only for nocturnal rodents where diet is only one source of vitamin D. In human, from evolutionary point of view, the skin production of vitamin D is its major source and does not pass thought intestinal epithelium. Please discuss.

5.       HCT 116 cell line was isolated from the colon of an adult male with colon cancer, thus may not represent normal intestinal epithelium.

6.        

Minor comments:

1.       Abstract and text, please be consistent, maybe CYP3As or cytochromes CYP3A (or CYP3A genes/enzymes) when you write about more than one.

2.       Line 365-67. Please correct the sentence something is wrong.

3.       Line 274-376. Are you sure that CYP24a1 55kDa protein is signal peptide of 35 aa?

4.        Please note that qPCR primers design to one exon only may also detect traces of DNA, not mRNA of CYP24A1 even if DNA treatment was performed.

5.       You can also mention that other CYPs like CYP11A1 is also capable to metabolize vitamin D and generate whole spectrum of active derivatives (Int J Mol Sci. 2018 Oct 8;19(10):3072, Dermatoendocrinol. 2013 Jan 1;5(1):7-19, Clin Chem Lab Med. 2021 May 20;59(10):1642-1652).

Comments on the Quality of English Language

As above

Author Response

Reviewer 3 Report (New Reviewer)

Comments and Suggestions for Authors

The study entitled "The role of intestinal Cytochrome P450s in vitamin D metabolism" by Minori Uga and colleagues investigated the catabolic role of CYP24A1 and CYP3A on 1,25(OH)2D3 bioactivity in vitro and the role of the intestine as a vitamin D target and metabolizing tissue in vivo. Here, the authors showed that 1,25(OH)2D3-dependent transcriptional activity and TRPV6 gene expression were reduced in FLAG-tagged CY3A or CYP24A1-transfected HEK293 cells, demonstrating the catabolic capacity for 1,25(OH)2D3 of both enzymes. Furthermore, this effect was observed only for full-length CYP24A1 and not for N-del CYP24A1, suggesting the importance of the cellular localization of CYP24A1 in the mitochondria to mediate 1,25(OH)2D3 hydroxylation. Furthermore, the authors showed that the mouse CYP3A4 ortholog CYP3a11 is mainly expressed in the intestine and liver, whereas Cyp3a13 is mainly expressed in the intestine. Cyp24a1 expression was only found at relevant levels in the kidney of mice. Interestingly, 1,25(OH)2D3 strongly increased Cyp24a1 gene and protein expression in the mitochondrial fraction of proximal and distal intestinal tissue of mice, whereas Cyp3a11 and Cyp3a13 were unaffected. Finally, the authors observed that 1,25(OH)2D3-mediated intestinal regulation of Cyp24a1 gene and protein expression was dependent on VDR and resulted in increased 25(OH)D3 catabolism in control compared to VDR-vKO mice. Thus, the results of this interesting study provide evidence for the bioactivity of CYP3A and CYP24A1 to metabolize 1,25(OH)2D3 for its degradation in intestinal and renal cells and that 1,25(OH)2D3 regulates Cyp24a1 expression in intestinal and renal tissues in a VDR-dependent manner. Finally, the data provide evidence that the intestine is not only a target but also a metabolizing tissue of 1,25(OH)2D3.

However, there are still some limitations, open questions, and areas for improvement that need to be addressed in a revision prior to publication.

1)     The main conclusion of the publication, that the intestine is also involved in the catabolism of vitamin D derivatives, is not entirely conclusive.

The mRNA levels appear to be much more tightly regulated (>100,000-fold) than the protein level of Cyp24a1 after treatment with 1,25D in the proximal and distal intestinal tissue of control mice. Similarly, the mRNA level of Cyp24a1 is also upregulated in the kidney, the main tissue of 1,25D degradation.

a)      The authors should therefore also determine the amount of Cyp24a1 protein in the kidneys of the animals of this experiment (both genotypes) after 1,25D or control treatment to show that the Cyp24a1-upregulation in the intestine is stronger at the protein level than in the kidney in order to be able to say that the intestine could be relevantly involved in 1,25D-degradation.

b)     Furthermore, a densitometric analysis of the Western blot data should be carried out and the intensities between the interventions and genotypes in the proximal and distal intestine and the added kidney should be determined and statistically compared.

c)      It is also questionable why only 3 of the 6 mice of each group were examined and shown in the study? All 6 animals should be evaluated in the densitometric analysis of intestinal and renal Cyp24a1 protein levels.

d)     Since 1,25D is the main bioactive vitamin D metabolite and substrate of Cyp24a1, the authors should measure plasma 1,25(OH)2D3 levels in the mice after 1,25D-treatment and if they are able 1,24,25(OH)3D3 to show that the increased intestinal Cyp24a1 is able to metabolize the oral administrated 1,25D. It seems more logical that the orally administered 1,25D is more efficiently and rapidly degraded in the control mice and therefore should be used as the reference metabolite to determine the hypothesis instead of 25(OH)D (that more reflects the supply status) , especially since it appears that the control mouse group with 1,25D treatment generally had higher 25(OH)D levels and therefore the elevated 24,25(OH)2D3 levels may have occurred indirectly. A more efficient hydroxylation of 1,25(OH)2D3 to 1,24,25(OH)3D3 and thus lower 1,25(OH)2D3 but higher 1,24,25(OH)3D3 plasma levels in the control mice compared to the VDR-vKO after administration of 1,25D would therefore provide better evidence for the hypothesis of intestinal 1,25D metabolism in this work and should therefore be investigated.

2)     Densitometric analysis of the western blot data should also be conducted and added in Fig. 4 for mitochondrial vs cytoplasmic Cyp24a1

3)     It is unclear which mouse or genotype was used under section 3.4 for the analysis of Cyp3a and Cyp24a1 expression. This information should be added.  

4)     It is unclear why there are different n-numbers used for the analysis. Why is there n=3-9 indicated in Fig. 1c. Why weren't 3 or 9 experiments carried out equally for all settings?

5)     Why were different 1,25D concentrations used by the authors in section 3.1 ?

6)     The discussion should take up the order of the results section or the results should be rearranged accordingly. The order of the data in the discussion appears more coherent than that of the current results section.

7)     Since multiple group comparisons of more than 2 groups were also carried out using ANOVA, the post-hoc test used should also be stated in addition to the ANOVA in the methods.

8)     Information’s of sex and age of the used mice in the experiments should be given in the method section

9)     The kDa size should be added in the figures (1-5) for all detected proteins, not only for Cyp24a1

Comments on the Quality of English Language

1)     Some of the headings are not clearly formulated:

a.      3.1 CYP3A or CYP24A1 on vitamin D activity in human colon cell

Suggestion: Effects of CYP3A or CYP24A1 on 1,25(OH)2D3 activity in human colon cells

b.      3.2. Functional effects of CYP24A1 on vitamin D activity

Suggestion: Functional effects of CYP24A1 on 1,25(OH)2D3 activity

c.      3.3 Cyp3a or Cyp24a1 expression in mouse

 Suggestion: Tissue expression of Cyp3a and Cyp24a1 in mice

d.      3.4 Regulation of CYP3a or Cyp24a1 in intestine of mouse

  Suggestion: 1,25(OH)2D3-dependent regulation of Cyp3a or Cyp24a1 in the mouse intestine

e.      3.5 Regulation Cyp24a1 in intestine-specific VDR-KO mouse

    Suggestion: 1,25(OH)2D3-dependent regulation of Cyp24a1 in intestine-specific VDR-KO mice

Round 2

Reviewer 3 Report (New Reviewer)

Comments and Suggestions for Authors

Thank you for the detailed reply and the adjustments to the manuscript. The paper is now ready for publication.

This manuscript is a resubmission of an earlier submission. The following is a list of the peer review reports and author responses from that submission.

Round 1

Reviewer 1 Report

Comments and Suggestions for Authors

In this manuscript the authors utilize a number of cell and animal based assays to assess the importance of Cyp3a enzymes and Cyp24a1 in the metabolism of vitamin D metabolites.

There is a lot of interesting data presented and the authors conclusions are generally supported by the data.  However, some important issues need to be addressed:

(1) The novel aspect of this paper is the potential finding that Cyp24a1 has a truncated form whose level is increased by 1,25 D treatment.  However, this novel finding requires additional validation.  If correct, the authors are challenging many aspects of what we think we know about CYp24a1, the most critical being that it is a predominantly mitochondrial protein that can metabolize both 25OHD and 1,25 D.    First, the authors need to repeat this with another antibody, since the GeneTex polyclonal antibody used by the authors has been discontinued. Second, the lower band that the authors propose is a truncated version of Cyp24a1 must be isolated and sequenced to verify it's identity.  Third, the authors must demonstrate that the lower MW band is concentrated in the mitochondria by showing it at high concentration in a mitochondrial preparation and low concentration in a cytosolic preparation.  In addition, if it turns out that this is indeed a N-terminal truncated version of the protein, the authors need to look at the crystal structure of the protein and propose how a mitochondrial localization signal would contribute to substrate specificity (i.e. as a potential mechanism to explain why 1,25(OH)2 D is not recognized by the truncated form they generated).

(2) The cell based work in Figure 1 suggests that Cyp3a enzymes can inactivate 1,25 D in a variety of cell types but Figure 3 indicates that the mouse version is not a vitamin D target genes.  However, other research indicates that both the mouse and human Cyp3a forms are induced by 1,25 D treatment in the intestine (see review by Qin, Wang (2019) Acta Pharm Sin B 9:1087; PMC6900549).  The authors should elaborate on this distinction.

(3) In figure 1, the individual Cyp's are examined.  Have the authors combined Cyp3a and Cyp24a1 to see if the biological effect of 1,25 D can be completely eliminated?  

(4) In the discussion on line 450 and the authors state that the regulatory mechanisms controlling the small intestine regulatory enzymes in the small intestine are unknown.  However, Lee et al. PMC4505063 and Aita et al. PMC9358460 have both published VDR ChIP-seq data on mouse small intestine that reveals the gene regulatory landscape of the Cyp24a1, Cyp3a11, and Cyp3a13 genes.  All of them have VDR binding sites induced by 1,25 D treatment.

(5) The authors conclude that their studies support the idea that the intestine is a vitamin D-metabolizing tissue.  While their data are consistent with this interpretation, the data do not directly test this with in vivo data.  Until they generate in vivo data showing some of the same things as their in vitro data, their conclusions should be stated more cautiously.

(6) Would the authors clarify more about how they designated proximal/distal intestine?  Where they using mucosal scrapings?  what length of segment was used and where exactly did the segment come from?  (e.g. 5 cm proximal to the pyloric sphincter)

Reviewer 2 Report

Comments and Suggestions for Authors

In this manuscript, the authors have explored the role of CYP enzymes, specifically CYP3A and CYP24A1, in the metabolism of vitamin D in the small intestine. The study demonstrates that these enzymes significantly reduce the activity of 1,25-dihydroxyvitamin D3 [1,25(OH)2D3], the active form of vitamin D. Moreover, in mice, CYP24A1 is induced by 1,25(OH)2D3 in the intestine, leading to the production of both full-length and truncated forms. The truncated form, found exclusively in intestinal mitochondria, appears to hydroxylate 25OHD3 but not 1,25(OH)2D3. The findings suggest a nuanced regulation of vitamin D metabolism in the small intestine but following points need to be carefully considered;

How does the gain-of-function mutation in CYP3A4 relate to vitamin D-dependent rickets, type III?

What is the significance of the expression of both full-length and truncated forms of CYP24A1 in response to 1,25(OH)2D3 in mice?

How does the study propose the small intestine's role as a newly-recognized vitamin D-metabolizing tissue?

What are the potential implications of the VDR-dependent nature of the truncated Cyp24a1 protein observed in the intestine-specific VDR-KO mouse model?
